# SurgiCal Obesity Treatment Study (SCOTS): a prospective, observational cohort study on health and socioeconomic burden in treatment-seeking individuals with severe obesity in Scotland, UK

Ruth M Mackenzie,[1] Nicola Greenlaw,[2] Abdulmajid Ali,[3] Duff Bruce,[4] Julie Bruce [ID],[5] Eleanor Grieve,[6] Mike Lean [ID],[7] Robert Lindsay,[1] Naveed Sattar [ID],[8] Sally Stewart [ID],[9] Ian Ford,[2] Jennifer Logue [ID],[10] on behalf of the SCOTS investigators

For numbered affiliations see end of article.

**Correspondence to**
Dr Jennifer Logue;
j.logue1@lancaster.ac.uk

## ABSTRACT

**Objectives** There is a lack of evidence to inform the delivery and follow-up of bariatric surgery for people with severe obesity. The SurgiCal Obesity Treatment Study (SCOTS) is a national longitudinal cohort of people undergoing bariatric surgery. Here, we describe characteristics of the recruited SCOTS cohort, and the relationship between health and socioeconomic status with body mass index (BMI) and age.

**Participants/Methods** 445 participants scheduled for bariatric surgery at any of 14 centres in Scotland, UK, were recruited between 2013 and 2016 for this longitudinal cohort study (1 withdrawal); 249 completed health-related preoperative patient-reported outcome measures. Regression models were used to estimate the effect of a 10-unit increase in age or BMI, adjusting for sex, smoking and socioeconomic status.

**Results** Mean age was 46 years and median BMI was 47 kg/m². For each 10 kg/m² higher BMI, there was a change of −5.2 (95% CI −6.9 to −3.5; p<0.0001) in Rand 12-item Short Form Survey Physical Component Summary (SF-12 PCS), −0.1 (95% CI −0.2 to −0.1; p<0.0001) in EuroQoL 5-level EQ-5D version index score and 14.2 (95% CI 10.7 to 17.7; p<0.0001) in Impact of Weight on Quality of Life-Lite Physical Function Score. We observed a 3.1 times higher use of specialist aids and equipment at home (OR: 3.1, 95% CI 1.9 to 5.0; p<0.0001). Broadly, similar results were seen for each 10-year higher age, including a change of −2.1 (95% CI −3.7 to −0.5; p<0.01) in SF-12 PCS.

**Conclusions** A higher BMI combined with older age is associated with poor physical functioning and quality of life in people seeking bariatric surgery treatment. Policy-makers must consider the health and care needs of these individuals and invest to provide increased access to effective weight management.

**Trial registration number** ISRCTN47072588.

## STRENGTHS AND LIMITATIONS OF THIS STUDY

⇒ A limitation is that selection for bariatric surgery is often based on the presence of comorbidity; so these results, while applicable to a treatment-seeking population, may not be directly applicable to the population with severe obesity in general.

⇒ The fact that almost every bariatric surgery patient in Scotland was recruited is a major strength of this study and renders it more representative than many studies undertaken in the field.

⇒ A further strength is that the majority of questionnaires used were externally validated and wide ranging, containing a number of unique questions covering medical, social, psychological and physical domains.

## INTRODUCTION

Obesity (body mass index (BMI) ≥30 kg/m²) and, in particular, severe obesity (BMI ≥40 kg/m²) are associated with a variety of negative health outcomes, including increased risk of most major chronic diseases.[1] In recent years, severe obesity has emerged as a major public health concern with rates increasing rapidly in a number of countries across the world, including the USA where the prevalence of BMI >40 kg/m² rose by 70% between 2000 and 2010,[2] and around 7.7% of adults are now considered to have severe obesity.[3] Similarly, levels of severe obesity have risen in the UK with 2.9% of all adults in England[4] and 4% of all adults in Scotland now estimated to have a BMI ≥40 kg/m².[5]

As the prevalence of severe obesity rises, effective treatment is a priority. The efficacy

of bariatric surgery for significant long-term weight loss is well established.[6 7] However, for the UK National Health Service (NHS), like many other health systems, the commissioning of bariatric surgery is low priority. Indeed, despite obesity prevalence being among the highest in the European Union, the UK performs only 9 bariatric surgery procedures per 100 000 people,[8 9] while Sweden, a country with a similar health service but lower obesity prevalence, performs 70–80 procedures per 100 000 people.[10] In North America, the rate of surgery is around 40–50 per 100 000 people, with the majority of these operations performed in the USA.[8]

The low prioritisation of bariatric surgery within the UK and the strict criteria for access to surgery, including complex pre-surgical pathways and pre-surgical weight-loss requirements,[5] results in low numbers of individuals with severe obesity actually receiving surgery. Those receiving surgery generally do so after many years of alternative conservative interventions, at a point when their mean BMI is extremely high, at around 45 kg/m$^2$, and they are at a median age of 47 years.[11] To date, it is unclear how this delay in treatment impacts on health, physical functioning and quality of life.

The SurgiCal Obesity Treatment Study (SCOTS) is the first national epidemiological study established to investigate long-term outcomes following bariatric surgery. This cohort study collected clinical and patient-reported health outcomes from treatment-seeking individuals from across Scotland with severe obesity before they underwent bariatric surgery.[12] The aim of the present study is to describe the health-related characteristics of the recruited SCOTS cohort and to examine relationships between age, preoperative BMI and other health-related factors in our recruited sample of adults with severe obesity from across Scotland.

## PARTICIPANTS AND METHODS

This study was registered prospectively at the International Standard Randomised Controlled Trials Number registry. In reporting our findings, we have adhered to the STrengthening the Reporting of OBservational studies in Epidemiology Statement[13] (online supplemental table 1).

### Patient involvement

Patients identified via bariatric surgery peer support groups in Scotland were involved in the design and conduct of this research. During the protocol development stage, patients provided input with regard to data collection and defining research questions. In addition, methods of recruitment were informed by discussions with patients during two focus group sessions. Patients were subsequently involved in reviewing paperwork for recruitment and appeared in the recruitment video. There is also a patient member of the independent study steering committee and patients were invited to a meeting to discuss plans for dissemination of study results.

### Study design

SCOTS is a national, prospective, observational cohort study of adults aged over 16 years who were eligible for primary bariatric surgery in Scotland. Participants were recruited over a 4-year period from December 2013 to February 2017. Recruitment to the cohort has now closed, although follow-up will continue until October 2020. A detailed protocol for the SCOTS was published previously[12] and is included as online supplemental file 1, but was amended in 2016 to reflect a smaller cohort size and shorter follow-up. This was in response to changes in national service commissioning contributing to lower numbers of bariatric surgery which, in turn, affected recruitment potential and planned sample size. Any changes from the original protocol are outlined below.

### Participant and centre eligibility

As outlined in the protocol[12] (online supplemental file 1), patients scheduled to undergo a primary bariatric surgery procedure at any of the 10 NHS-funded or 4 private hospitals in Scotland providing this surgery were eligible for invitation to the study. Inclusion criteria were that patients were aged 16 years or over, and undergoing their first bariatric surgery procedure. Patients were required to have capacity to consent, provide written informed consent and be resident in Scotland. Patients who did not meet these criteria or who did not have English language skills to complete written questionnaires were ineligible[12] (online supplemental file 1).

### Recruitment procedures and consent

Patients were approached about the study at least 4 weeks prior to their primary bariatric surgical procedure by the clinical bariatric surgery team or by a research nurse in preoperative assessment clinics. Written signed consent for access to medical records, linkage of electronic health records and for postal questionnaires was obtained on a subsequent clinical visit.

### Data collection

Recruited participants were asked to complete questionnaires preoperatively and at 2 years and 3 years, postoperatively. Prior to receiving questionnaires, participants were made aware (via patient information leaflets) that the estimated time for completion of each questionnaire was 1 hour. The approximate time it would take to complete the questionnaire (1 hour) was also stated clearly on the first page of each questionnaire received by participants. Completion could be either by post or electronically via a secure link sent by email. Two reminders were sent by the participant's chosen method and a third reminder, if required, was sent by post to all participants. No further strategy was used after three reminders.

Baseline preoperative questionnaires collected health-related information, including weight, medical history, smoking status, alcohol use, gastrointestinal symptoms,

urological health, depression, anxiety, health-related quality of life (HRQoL) and obesity-specific quality of life (O-QoL), life optimism, physical activity, healthcare utilisation, employment and social security. Comorbidity was assessed by self-report using a questionnaire designed specifically for this study (online supplemental file 2). Gastrointestinal symptoms were evaluated using a questionnaire developed for the REFLUX trial.[14] Urological health was assessed using the International Prostate Symptom Score (IPSS)[15] and the International Consultation on Incontinence Questionnaire-Urinary Incontinence Short Form (ICIQ-UI SF) wherein a score ≥6 indicates moderate incontinence.[16] Female reproductive health data were obtained using a modified version of the questionnaire developed for the Longitudinal Assessment of Bariatric Surgery-2 study.[17] Information on male erectile dysfunction was obtained using a modified version of the questionnaire developed for the Massachusetts Male Ageing Study.[18] Anxiety and depression were assessed using the 7-item Generalised Anxiety Disorder Assessment (GAD-7)[19] and Patient Health Questionnaire-9 (PHQ-9)[20] instruments, respectively. A PHQ-9 score ≥10 is indicative of moderate to severe depression, while a GAD-7 score ≥6 is reflective of moderate to severe anxiety.[19 20] Smoking status was ascertained using a questionnaire specifically developed for this study (online supplemental file 2). Alcohol use was determined using a modified version of the Alcohol Use Disorders Identification Test.[21] HRQoL was assessed using the Rand 12-item Short Form Survey (SF-12)[22] and EuroQoL 5-level EQ-5D version (EQ-5D-5L)[23 24] instruments while O-QoL was assessed using the Impact of Weight on Quality of Life-Lite (IWQOL-Lite) questionnaire.[25] Standardised scoring was used when interpreting IWQOL-Lite questionnaires.[26] Life optimism was determined using a modified version of the Life Orientation Test, wherein a score range of 0–13 reflects low optimism (high pessimism).[27] Data on physical activity were obtained using the International Physical Activity Questionnaire (IPAQ) Short Form.[28] Information on participants' employment, social security status and healthcare utilisation was obtained using questionnaires specifically developed for this study (online supplemental file 2).

Each participant's quintile of the Scottish Index of Multiple Deprivation (SIMD), an area-based measure of socioeconomic status,[29] was derived from their postcode. Combining a number of indicators of socioeconomic status across seven domains, the SIMD provides a relative measure of deprivation which can be used to compare data zones by ranking them from most to least deprived. The seven domains include income, employment, health, education, skills and training, housing, geographic access and crime.[29]

Height and weight data were collected by clinical staff at recruitment allowing BMI to be calculated.

We herein report baseline data recorded during the recruitment visit for the whole recruited cohort before bariatric surgery, and patient-reported outcomes for the subset completing baseline questionnaires.

## Statistical analyses

Continuous data are reported as means and SD or medians and lower (Q1) and upper (Q3) quartiles depending on data distribution, and counts and percentages are reported for categorical data. Age and BMI data were categorised a priori; BMI ($kg/m^2$) is reported as 5-unit bands (<40, 40–44, 45–49, 50–54 and ≥55) and age was categorised using 10-year age bands (<35 years, 35–44 years, 45–49 years, 50–54 years and ≥55 years). Other baseline demographics are summarised by group.

Linear regression models were used to examine continuous quality of life measures and logistic regression for the binary 'need for specialist aids' and 'equipment in the home to assist with daily living'. Models were used to estimate the effect of a 10-unit increase in age or BMI in (1) an unadjusted model and (2) in a model adjusted for sex, SIMD, smoking status and BMI or age, respectively, as these factors may be associated with obesity and related comorbidity. Regression model effect estimates, or ORs, and corresponding 95% CIs and associated p values are provided. Data were analysed as available, without any imputation for missing data. All analyses were performed using SAS (V.9.3).

## RESULTS

Over the 3-year recruitment period, a total of 548 patients were approached and screened for eligibility to participate. Of these, 103/548 (19%) were excluded or declined to participate (figure 1). We recruited 445/548 (81%) participants, but one participant withdrew consent leaving a recruited sample of 444 (81%). Of the recruited sample, 413/444 (93%) consented to data linkage and questionnaire follow-up, while 31/444 (7%) consented to data linkage only. Of these 413 participants, a total of 164/413 (40%) were not included in the subsequent analysis: 129 did not return a baseline questionnaire and 35 had bariatric surgery before their baseline patient-reported outcome measures (PROMs) questionnaires were completed. Of the 129 participants who did not return baseline questionnaires, 84/129 (65%) progressed to surgery, 43/129 (33%) did not progress to surgery and the status of 2/129 (2%) was unknown. Completed preoperative baseline PROMs data for 249/413 participants (60% of those consented) were available for analysis (figure 1).

### Characteristics of recruited and analysed sample

Demographic data are summarised in table 1. Participant characteristics were similar between the total recruited sample (n=444) and the analysed subset (n=249) with completed PROMS before bariatric surgery (table 1). Mean age was 46 years (SD: 9.1 years) with a higher proportion of women than men (71% vs 29%). Half of recruited participants were aged 35–49 years, with one-third being over 50 years. The median BMI was 47 $kg/m^2$ (Q1: 43; Q3: 54) with more than 21% having a BMI of ≥55 $kg/m^2$. Over half of the participants (55%)

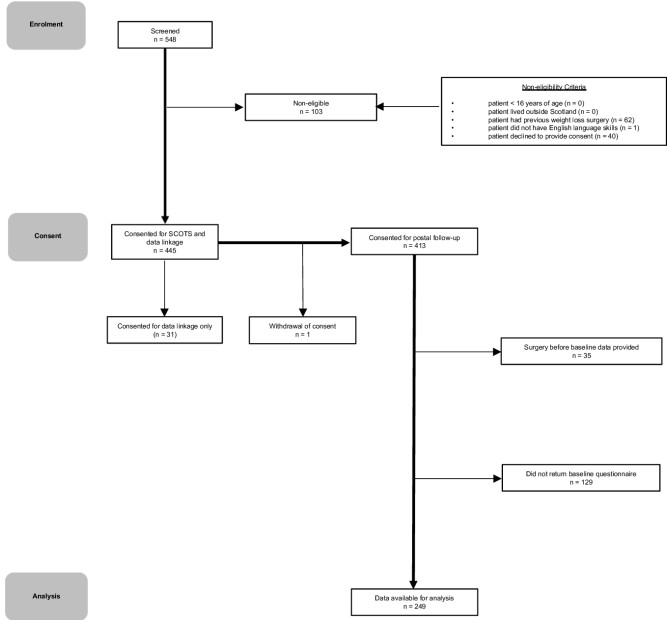

**Figure 1** Screening, consent and follow-up. 548 patients were screened for inclusion in the SCOTS. Of these 548 patients, 103 were non-eligible (62 patients had undergone previous bariatric surgery and 1 patient lacked the necessary English language skills) and 40 were eligible but did not consent. 31 patients consented to only part 1 of the study (data linkage), while 413 consented to both part 1 and part 2 (completion of baseline PROMs). 249 baseline PROMs were available for analysis. PROMs, patient-reported outcome measures; SCOTS, SurgiCal Obesity Treatment Study.

lived in areas of high socioeconomic deprivation (SIMD quintiles 1 and 2). There were no statistically significant differences between the analysed subset (n=249) and the non-analysed subset (n=195, data not shown).

## Comorbidities

For the analysed sample (n=249), self-reported medical comorbidities, and physical, mental and functional measures are presented in table 2. Over 40% reported having at least one of hypertension, type 2 diabetes (T2D), back problems, anxiety/depression and gastro-oesophageal reflux. Over 60% of the sample reported more than three comorbidities. Over 40% of male participants reported erectile dysfunction, while one-third of males described urinary incontinence. Half of female participants reported urinary incontinence. Mean depression scores reflected mild depression, although 44% of participants had scores indicating moderate to severe depression. Anxiety scores for all participants were indicative of mild anxiety (median: 5.0), with half of participants having scores indicative of moderate to severe anxiety. The mean life optimism score for participants was reflective of low optimism (high pessimism). Very few participants smoked (5%) and, on average, alcohol consumption was moderate.

## Health and obesity-related quality of life

Mean SF-12 Physical Component Summary (PCS) and Mental Component Summary (MCS) scores were

low—PCS: 37.0 (11.4) and MCS: 45.5 (10.3). The median EQ-5D-5L score of sample participants was 0.6 (0.3; 0.8), while the mean EQ-5D-5L Visual Analogue Scale was 55.3 (22.1). Participants had a mean IWQOL-Lite Physical Function score of 56.9 (25.4) and a mean total score of 58.5 (21.7) (table 2), where an increase in IWQOL-Lite score indicates a worsening in quality of life.

## Physical activity

Over 80% of SCOTS participants reported undertaking at least 10 min of either walking, moderate or vigorous activity in the last 7 days and the median IPAQ score for the sample was 720.0 metabolic equivalent of task (MET) min/week. Almost one-third (29%) of participants reported using aids or specialist equipment to assist with their daily activities in the home (table 2).

## Comorbidity by BMI and age

Comorbidity data are presented by BMI group and age group in online supplemental tables 2 and 3, respectively. In order to further investigate the associations between BMI and age on physical, mental and functional measures, and healthcare utilisation within the SCOTS population, regression analyses were performed (table 3). There was no significant correlation between BMI and age (correlation: 0.0075, p=0.9061). Higher BMI values and higher ages were negatively associated with physical, but not mental, HRQoL scores (table 3). For each 10 kg/m$^2$ higher BMI, there was a change of −5.2 (95% CI −6.9 to −3.5; p<0.0001) in SF-12 PCS, −0.1 (95% CI −0.2 to −0.1; p<0.0001) in EQ-5D-5L score and 14.2 (95% CI 10.7 to 17.7; p<0.0001) in IWQOL-Lite Physical Function score (where an increase in score indicates a worsening). We observed a 3.1 times higher use of specialist aids and equipment in the home (OR: 3.1, 95% CI 1.9 to 5.0; p<0.0001), adjusting for age, sex, smoking and socioeconomic deprivation. For each 10-year higher age, there was a change of −2.1 (95% CI −3.7 to −0.5; p<0.01) in SF-12 PCS score, −0.1 (95% CI −0.1 to 0.0; p<0.01) in EQ-5D-5L score and 5.01 (95% CI 1.8 to 8.3; p<0.01) in IWQOL-Lite Physical Function score, and a 3.1 (OR 3.1, 95% CI 1.9 to 5.0; p<0.0001) times higher use of specialist aids or equipment in the home, adjusting for BMI, sex, smoking and socioeconomic status.

Interactions were explored between smoking and both age and BMI (with smoking as a two-level variable (smoked or never smoked) due to small numbers in the current smoker group) and a borderline significant interaction between age and smoking status was observed (online supplemental table 4). Further exploration in the subpopulations of smokers (current or former) and those who had never smoked revealed a significant effect of age on the use of specialist aids or equipment in the home in both subpopulations, but the OR suggests a trend toward a slightly larger odds in those participants who had never smoked (online supplemental table 4). No significant effect of age or BMI on moderate to severe depression (PHQ-9) was observed in either the unadjusted or

Table 1  Baseline sociodemographic characteristics of recruited (N=444) and analysed (N=249) samples of SCOTS participants

| Characteristic | | Recruited sample N=444 | Analysed sample* N=249 |
|---|---|---|---|
| Sex, N (%) | Male | 123 (27.7) | 72 (28.9) |
| | Female | 321 (72.3) | 177 (71.1) |
| | Missing | 0 | 0 |
| Age (years) | Mean (SD) | 46.2 (9.1) | 45.9 (9.1) |
| | Missing | 0 | 0 |
| Age group, N (%) | <35 years | 61 (13.7) | 36 (14.5) |
| | 35–44 years | 116 (26.1) | 63 (25.3) |
| | 45–49 years | 109 (24.5) | 63 (25.3) |
| | 50–54 years | 79 (17.8) | 43 (17.3) |
| | 55+ years | 79 (17.8) | 44 (17.7) |
| | Missing | 0 | 0 |
| BMI (kg/m$^2$) | Median (Q1; Q3) | 47.2 (42.7; 53.6) | 47.6 (42.8; 53.8) |
| | Missing | 1 | 0 |
| BMI group, N (%) | <40 | 52 (11.7) | 24 (9.6) |
| | 40–44 | 115 (26.0) | 64 (25.7) |
| | 45–49 | 116 (26.2) | 64 (25.7) |
| | 50–54 | 71 (16.0) | 44 (17.7) |
| | 55+ | 89 (20.1) | 53 (21.3) |
| | Missing | 1 | 0 |
| SIMD quintile, N (%) | 1 (most deprived) | 135 (30.5) | 70 (28.3) |
| | 2 | 108 (24.4) | 65 (26.3) |
| | 3 | 84 (19.0) | 51 (20.6) |
| | 4 | 68 (15.4) | 34 (13.8) |
| | 5 (least deprived) | 47 (10.6) | 27 (10.9) |
| | Missing | 2 | 2 |
| Marital status, N (%) | Married/civil partnership/co-habiting | Not collected | 155 (63) |
| | Single/separated/divorced/widowed | | 91 (37) |
| | Missing | | 3 |
| Ethnic group, N (%) | White | Not collected | 243 (97.6) |
| | Mixed | | 4 (1.6) |
| | Asian/Asian Scottish/Asian British | | 1 (0.4) |
| | African Caribbean/Black | | 1 (0.4) |
| | Other | | 0 (0.0) |
| | Missing | | 0 |
| Education, N (%) | School only | Not collected | 58 (23.5) |
| | Formal qualifications through training at work | | 54 (21.9) |
| | Qualification (other than a degree from college or university) | | 64 (25.9) |
| | Degree from college or university | | 71 (28.7) |
| | Missing | | 2 |

Continued

**Table 1** Continued

| Characteristic | | Recruited sample N=444 | Analysed sample* N=249 |
|---|---|---|---|
| Current employment status, N (%) | Working full time | Not collected | 124 (50.0) |
| | Working part time | | 24 (9.7) |
| | Unable to work because of illness or disability | | 64 (25.8) |
| | Student/unemployed and seeking employment/ unemployed and not seeking employment/ carer/other | | 36 (14.5) |
| | Missing | | 1 |

Working full time: ≥ 30 hours per week; working part time: < 30 hours per week. Percentages may not total 100% in all cases due to rounding.
*Participants who returned baseline questionnaires prior to their bariatric surgery are included in the analysed sample.
BMI, body mass index; SCOTS, SurgiCal Obesity Treatment Study.

adjusted models. However, on extending that model to include the interactions between smoking and each of age and BMI, we observe a significant interaction between BMI and smoking status. Considering the subpopulations of smokers and those who had never smoked, there was no significant effect of BMI on smokers, but there was a significant effect of BMI in those who never smoked, with increasing BMI having increased odds of moderate to severe depression (online supplemental table 4).

With regard to medical comorbidities, as listed in online supplemental table 5, higher BMI had a significant association with higher prevalence of asthma in the SCOTS population, while older age was associated with higher prevalence of hypertension, arthritis and sleep apnoea (online supplemental table 5).

## DISCUSSION

Despite escalating levels of severe obesity in Western society and the concomitant increase in bariatric surgical procedures being performed in some countries,[30] there is a dearth of information on the health status of people living with severe obesity. This is the first report from the national Scottish cohort study of people seeking surgical treatment for severe obesity. We recruited 444 adults from 14 centres across Scotland over a 3-year period, including all NHS centres and major private hospitals undertaking bariatric surgery. We found that a higher BMI and older age were associated with decreased physical quality of life, increased use of specialist aids and equipment in the home and a high prevalence of comorbidities.

There has been a significant increase in the prevalence of a BMI $\geq 40 \, kg/m^2$ in recent decades; in Scotland, obesity prevalence has trebled in women aged 16–64 years since 1995. However, it is hard to assess the global increase due to lack of reporting of a BMI $\geq 40 \, kg/m^2$ in national health survey data.[31] While it is known that healthcare resource use increases in people with a BMI $\geq 30 \, kg/m^2$, with service use estimated to be over 25% higher than for those with a BMI in the normal weight range,[32] few data exist for those with a BMI $\geq 40 \, kg/m^2$. In 2016, the Global BMI Mortality

Collaboration[33] conducted an individual-participant-data meta-analysis of 239 prospective studies and found a 2.8 times increased risk of all cause mortality for people with a BMI of 40–60 $kg/m^2$. Grieve et al[34] conducted a systematic review, which focused on the economic cost of severe obesity (BMI $\geq 40 \, kg/m^2$) and found limited literature describing increased prescribing, outpatient utilisation and intensive care admission, and hospital length of stays during critical illness. However, in both studies, there was no disaggregation of BMI beyond >40 $kg/m^2$ meaning that the health consequences of severe obesity are not yet fully described.

There has been extensive research on the relationship between HRQoL and obesity.[35] Ul-Haq et al[36] performed a meta-analysis of 8 studies (43 086 participants) and found that physical quality of life, measured by the Rand 36-item Short Form Health Survey, was reduced by 9.7 points in those with a BMI of 40 $kg/m^2$ compared with those with a BMI in the normal range, although, again, there was no disaggregation above BMI 40 $kg/m^2$. Van Nunen et al[37] performed a meta-analysis to compare the general, non-treatment-seeking population to patients within weight management programmes and those seeking bariatric surgery. They found that those seeking surgical treatment reported the most severely reduced HRQoL, perhaps reflecting their reasons for seeking definitive surgical treatment. Our cohort of treatment-seeking individuals who completed a rich battery of patient-reported measures provides data to show that health-related and obesity-related quality of life of those with the highest body mass is extremely poor and this is compounded by increasing age. Quality of life scores of SCOTS participants in both the upper BMI ($\geq 55 \, kg/m^2$) and older age ($\geq 55$ years) groups included physical scores comparable with those reported by cancer patients receiving palliative care,[38] patients with chronic heart failure expressing end of life preferences[39] and patients with end stage kidney disease.[40] Furthermore, patients with severe chronic obstructive pulmonary disease report higher quality of life scores, indicating a better quality of life, than our

**Table 2**  Preoperative health-related characteristics of SCOTS participants undergoing bariatric surgery (N=249)

| | | N=249<br>N (%) | Missing<br>N (%) |
|---|---|---|---|
| Comorbidity, self-report | DVT | 8 (3.2) | 0 |
| | Pulmonary embolism | 4 (1.6) | 0 |
| | Hypertension | 107 (43.0) | 0 |
| | T2D | 124 (49.8) | 0 |
| | Angina/heart attack | 17 (6.8) | 0 |
| | Heart failure | 2 (0.8) | 0 |
| | Stroke/ministroke | 6 (2.4) | 0 |
| | Arthritis | 73 (29.3) | 0 |
| | Back problems | 115 (46.2) | 0 |
| | Chronic bronchitis | 4 (1.6) | 0 |
| | Eczema/psoriasis | 33 (13.3) | 0 |
| | Asthma | 70 (28.1) | 0 |
| | Thyroid problems | 32 (12.9) | 0 |
| | Migraine | 49 (19.7) | 0 |
| | Anxiety/depression | 114 (45.8) | 0 |
| | Kidney disease | 7 (2.8) | 0 |
| | Liver disease | 2 (0.8) | 0 |
| | Cancer | 4 (1.6) | 0 |
| | Irritable bowel syndrome | 44 (17.7) | 0 |
| | Sleep apnoea | 66 (26.5) | 0 |
| | CVD | 20 (8.0) | 0 |
| N (%), self-reported comorbidities | None | 9 (3.6) | 0 |
| | 1–2 | 80 (32.1) | 0 |
| | ≥3 | 160 (64.3) | 0 |
| Gastro-oesophageal reflux | Yes | 97 (40.4) | 9 (3.6) |
| Female reproductive health, N=75* | Mean (SD) age in years, last natural menstrual period | 39.4 (10.9) | 4 (5.3) |
| Female reproductive health, N=177 | PCOS, N (%) | 28 (16.8) | 10 (5.6) |
| Male reproductive health, N=72 | Impotence, N (%) | 28 (41.2) | 4 (5.6) |
| | IPSS score ≥8, N (%) | 34 (47.9) | 1 (1.4) |
| Incontinence | Median (Q1; Q3) ICIQ-UI SF score | 4 (0.0; 10.0) | 10 (4.0) |
| | ICIQ-UI SF score ≥6 | 105 (43.9) | 10 (4.0) |
| Incontinence, females, N=177 | ICIQ-UI SF score ≥6, N (%) | 83 (49.4) | 9 (5.1) |
| Incontinence, males, N=72 | ICIQ-UI SF score ≥6, N (%) | 22 (31.0) | 1 (1.4) |
| Depression | Mean (SD) PHQ-9 score | 9.6 (6.3) | 5 (2.0) |
| | N (%) PHQ-9 score ≥10 | 107 (43.9) | 5 (2.0) |
| Anxiety | Median (Q1; Q3) GAD-7 | 5 (2.0; 9.0) | 6 (2.4) |
| | N (%) GAD-7 score ≥6 | 114 (46.9) | 6 (2.4) |
| Smoking status | Current | 13 (5.4) | 9 (3.6) |
| | Former | 105 (43.8) | |
| | Never | 122 (50.8) | |
| Alcohol use | Median (Q1; Q3) AUDIT | 3 (1.0; 6.0) | 20 (8.0) |

**Table 2** Continued

|  |  | N=249 N (%) | Missing N (%) |
|---|---|---|---|
| Quality of life |  |  |  |
| SF-12 | Mean (SD) PCS | 37 (11.4) | 13 (5.2) |
|  | Mean (SD) MCS | 45.5 (10.3) | 13 (5.2) |
| EQ-5D-5L | Median (Q1; Q3) | 0.6 (0.3; 0.8) | 12 (4.8) |
|  | Mean (SD) VAS | 55.3 (22.1) | 12 (4.8) |
| IWQOL-Lite (standardised scoring) | Mean (SD) physical function | 56.9 (25.4) | 6 (2.4) |
|  | Mean (SD) self-esteem | 70.7 (27.1) | 7 (2.8) |
|  | Mean (SD) sexual life | 57.1 (31.7) | 18 (7.2) |
|  | Mean (SD) public distress | 58.1 (27.2) | 6 (2.4) |
|  | Mean (SD) work | 43.6 (29.2) | 13 (5.2) |
|  | Mean (SD) total score | 58.5 (21.7) | 7 (2.8) |
| Life optimism | Mean (SD) LOT score | 13 (4.9) | 14 (5.6) |
| Physical activity | ≥1 walking, moderate or vigorous activity in last 7 days | 201 (83.4) | 8 (3.2) |
|  | Median (Q1; Q3) IPAQ score† | 720 (40.0; 1800.0) | 6 (3.0) |
| Healthcare utilisations | Using any aids or specialist equipment | 67 (28.9) | 17 (6.8) |
|  | Median (Q1; Q3) GP visits in last 3 months | 2 (1.0; 3.0) | 79 (31.7) |
|  | Median (Q1; Q3) visits to other health/social care providers in last 3 months | 3 (1.0; 5.0) | 73 (29.3) |
| Social security | Unable to work due to illness or disability | 64 (25.8) | 1 (0.4) |
|  | Receiving disability living allowance (caring) | 44 (18.6) | 13 (5.2) |
|  | Receiving disability living allowance (mobility) | 47 (19.9) | 13 (5.2) |

IPSS Scores: 0–7='mild symptoms'; 8–19='moderate symptoms'; 20–35='severe symptoms'.[15] ICIQ-UI SF score: ≥6='moderate incontinence'.[16] PHQ-9 scores: 0–4='minimal depression'; 5–9='mild depression'; 10–14='moderate depression'; 15–19='moderately severe depression'; 20–27='severe depression'.[20] GAD-7 scores: 0–5='mild anxiety'; 6–10='moderate anxiety'; 11–15='moderately severe anxiety'; 15–21='severe anxiety'.[19]
*75/177 (42%) female participants reported not menstruating in the last 12 months
†Metabolic equivalent of task min/week.
AUDIT, Alcohol Use Disorders Identification Test; CVD, cardiovascular disease; DVT, deep vein thrombosis; EQ-5D-5L, EuroQoL 5-level EQ-5D version; GAD-7, 7-item Generalised Anxiety Disorder Assessment; GP, general practitioner; ICIQ-UI SF, International Consultation on Incontinence Questionnaire-Urinary Incontinence Short Form; IPAQ, International Physical Activity Questionnaire; IPSS, International Prostate Symptom Score; IWQOL-Lite, Impact of Weight on Quality of Life-Lite; LOT, Life Orientation Test; MCS, Mental Component Summary; MET, Metabolic equivalent of task; PCOS, polycystic ovary syndrome; PCS, Physical Component Summary; PHQ-9, Patient Health Questionnaire-9; SCOTS, SurgiCal Obesity Treatment Study; T2D, type 2 diabetes; VAS, visual analogue scale.

cohort of treatment-seeking obese participants.[41] As far as we are aware, this is the first study to investigate physical and mental health in patients with severe obesity awaiting bariatric surgery with finer level consideration of a BMI up to ≥55 kg/m².

UK guidelines[42] currently indicate that bariatric surgery is a treatment option for those with a BMI ≥40 kg/m², or between 35 kg/m² and 40 kg/m² in the presence of other significant diseases (eg, T2D or hypertension), which could be improved if they lost weight. Non-surgical weight management must have been attempted but not resulted in clinically beneficial weight loss before surgery is indicated. However, baseline SCOTS data appear to suggest that the low prioritisation of bariatric surgery and a lengthy preoperative pathway in the UK is associated with surgical treatment being reserved for individuals at an older age with a very high BMI. Indeed, in 2018, the Global Registry initiative of the International Federation for the Surgery of Obesity and Metabolic Disorders (IFSO) reported a global median pre-bariatric surgery for those with a BMI of 41.7 kg/m²,[43] as compared with 47.6 kg/m² in the SCOTS cohort. Similarly, IFSO reported a median patient age of 42 years at the time of bariatric surgery,[43] as compared with a median age of 47 years for SCOTS participants. This combination of higher BMI and older age means that, at the time of surgery, Scottish

**Table 3** Association of age and BMI with QoL, use of specialist equipment in the home and social security

| QoL indicators and functional measures | Variable | Unadjusted models* Regression coefficient (95% CI)‡ | Adjusted models† Regression coefficient (95% CI)‡ |
|---|---|---|---|
| SF-12 PCS | BMI | −4.91 (−6.55 to 3.28) | −5.21 (−6.90 to 3.52) |
| | Age | −2.44 (−4.03 to 0.85) | −2.14 (−3.73 to 0.54) |
| SF-12 MCS | BMI | −0.49 (−2.07 to 1.09) | −0.40 (−2.06 to 1.26) |
| | Age | 0.42 (−1.04 to 1.87) | 0.68 (−0.89 to 2.25) |
| EQ-5D-5L score | BMI | −0.11 (−0.15 to 0.06) | −0.11 (−0.16 to 0.06) |
| | Age | −0.08 (−0.12 to 0.03) | −0.07 (−0.11 to 0.02) |
| EQ-5D-5L VAS | BMI | −7.08 (−10.38 to 3.79) | −7.73 (−11.10 to 4.36) |
| | Age | −2.19 (−5.30 to 0.92) | −0.65 (−3.77 to 2.46) |
| IWQOL-Lite physical function | BMI | 13.72 (10.28 to 17.17) | 14.20 (10.69 to 17.70) |
| | Age | 5.77 (2.28 to 9.26) | 5.01 (1.75 to 8.27) |
| IWQOL-Lite self-esteem | BMI | 5.09 (1.02 to 9.16) | 5.99 (1.86 to 10.12) |
| | Age | −4.74 (−8.51 to 0.97) | −5.11 (−8.96 to 1.25) |
| IWQOL-Lite sexual life | BMI | 5.56 (0.71 to 10.41) | 5.74 (0.82 to 10.67) |
| | Age | 3.86 (−0.75 to 8.47) | 3.01 (−1.71 to 7.73) |
| IWQOL-Lite public distress | BMI | 15.36 (11.72 to 19.00) | 16.07 (12.34 to 19.80) |
| | Age | −2.78 (−6.57 to 1.02) | −3.04 (−6.51 to 0.44) |
| IWQOL-Lite work | BMI | 9.54 (5.25 to 13.84) | 9.59 (5.16 to 14.02) |
| | Age | 1.95 (−2.25 to 6.16) | 1.04 (−3.15 to 5.24) |
| IWQOL total score | BMI | 10.31 (7.29 to 13.33) | 10.88 (7.79 to 13.97) |
| | Age | 1.20 (−1.84 to 4.24) | 0.55 (−2.33 to 3.43) |
| Use of aids or specialist equipment | BMI | 2.34 (1.62 to 3.39)* | 3.10 (1.94 to 4.95)* |
| | Age | 2.65 (1.76 to 4.00)* | 3.10 (1.94 to 4.95)* |
| Disability living allowance (caring) | BMI | 1.19 (0.82 to 1.73)* | 1.07 (0.72 to 1.59)* |
| | Age | 1.54 (1.03 to 2.31)* | 1.62 (1.08 to 2.46)* |
| Disability living allowance (mobility) | BMI | 1.18 (0.82 to 1.70)* | 1.13 (0.77 to 1.65)* |
| | Age | 1.64 (1.10 to 2.45)* | 1.62 (1.08 to 2.44)* |

N ≥10 for all indicators and measures for which regression analyses were performed.
*Unadjusted models, including only the effect of BMI (per 10 kg/m$^2$) or age (per 10 years) on QoL indicators and functional measures
†Adjusted models, including the effects of BMI (per 10 kg/m$^2$) and age (per 10 years) on QoL indicators and functional measures, after adjusting additionally for sex, SIMD and smoking status
‡Regression coefficient (95% CI) is estimate (95% CI) for results from the linear regression and OR (95% CI) for results from the logistic regression. OR results are indicated with an asterisk mark (*).
BMI, body mass index; EQ-5D-5L, EuroQoL 5-level EQ172 5D version; IWQOL-Lite, Impact of Weight on Quality of Life-Lite; MCS, Mental Component Summary; QoL, quality of life; SF-12 PCS, 12-item Short Form Survey Physical Component Summary; SIMD, Scottish Index of Multiple Deprivation; VAS, Visual Analogue Scale.

patients have high levels of comorbidity and poor physical functioning. Bariatric surgery is considered a highly cost-effective intervention.[44] However, the health economic models rely on data primarily from the US and Scandinavian studies,[45–49] where BMI and age at the time of surgery are lower than in the UK. A higher BMI and older age are risk factors for postoperative complications[50 51] and also associated with lower total weight loss.[52–54] T2D remission rates are negatively correlated with age.[55] As such, focusing bariatric surgery provision on those with older age and higher BMI may result in higher costs of surgery with increased length of hospital stay, higher rates of postoperative complications, lower overall weight loss

and lower rates of disease remission. Consequently, the impressive health benefits and resultant cost-savings of bariatric surgery observed in clinical trials and observational cohorts from other countries may not be fully realised for the UK/Scottish population.

The SCOTS dataset represents a unique and rich resource. A major strength of the study is its representativeness. Indeed, every clinical team providing publicly funded bariatric surgery in Scotland approached their patients for recruitment to the study, rendering it highly representative of the population in comparison to other studies undertaken in the field. However, the number of participants with valid baseline questionnaires was lower

than anticipated. In many cases, this could be attributed to the participant undergoing surgery before completing the questionnaire, or the participant leaving the bariatric surgery pathway before surgery. The overall length of the questionnaire may have also played a role. Participants living in the most deprived areas were well represented in our cohort and the mean quality of life findings were broadly similar to those of bariatric surgery cohorts from across the world.[56–58] A further strength of the study is that questionnaires were externally validated and wide ranging, containing a number of unique questions covering medical, social, psychological and physical functioning domains. This wide range of self-reported health measures will allow us to account for a range of potentially mediating and confounding factors in future analyses. In addition, we have revealed the extent of comorbidities, including musculoskeletal, urinary and mental health problems affecting people with severe obesity. Low numbers of some comorbidities meant that this could not be a focus of this analysis.

A limitation of this study is that selection for bariatric surgery is often based on the presence of comorbidity; so these results, while applicable to a treatment-seeking population, may not be directly applicable to the whole population with severe obesity in the wider society. While we will have access to medical records via electronic health record data linkage in follow-up, the current analyses are based on self-report of selected comorbidities. It is well known that self-reported weights are underreported, particularly by people with a very high BMI.[59] However, we are confident of the accuracy of weight and height as these data were collected in clinic during the recruitment visit. With regard to future analyses, follow-up of the SCOTS cohort is ongoing, which will allow us to report the longitudinal health trajectory after bariatric surgery. The electronic health record infrastructure in Scotland[60] will also permit us to study health outcomes of SCOTS participants across a range of clinical events and, ultimately, determine the effect of bariatric surgery and other factors associated with health outcomes.

Obesity is a multisystem disease which affects every facet of a person's life. Our data have shown that a higher BMI combined with older age is associated with very poor physical functioning, and health-related and obesity-related quality of life. Indeed, quality of life scores for those living with severe obesity in Scotland are akin to those seen in the end stage of diseases such as cancer and heart failure. The health consequences of severe obesity and the extent to which treatments such as bariatric surgery can improve these are not yet known. Researchers should ensure that they include people with severe obesity in population cohorts and treatment studies, and study the impact of severe obesity in more detail; there are substantial differences in the health status of those with a BMI above 50 kg/m$^2$ and those whose BMI is around 40 kg/m$^2$. Policy-makers should consider the health and care needs of the growing numbers of individuals living with obesity. There will be a considerable future demand for healthcare and services must be designed to accommodate the physical needs of the individuals. While primary prevention of obesity is clearly paramount to avoid more people developing such a debilitating, chronic condition, investment is urgently needed, both in the UK and globally, to provide increased access to bariatric surgery and other forms of effective weight management, directly targeting patient groups who will benefit from surgical intervention as early in the disease course as possible.

**Author affiliations**
[1]BHF Glasgow Cardiovascular Research Centre, Institute of Cardiovascular and Medical Sciences, University of Glasgow, Glasgow, UK
[2]Robertson Centre for Biostatistics, University of Glasgow, Glasgow, UK
[3]University Hospital Ayr, Ayr, UK
[4]NHS Grampian, Aberdeen, UK
[5]Warwick Clinical Trials Unit, University of Warwick, Coventry, UK
[6]Institute of Health and Wellbeing, University of Glasgow, Glasgow, UK
[7]School of Medicine, Dentistry and Nursing, University of Glasgow, Glasgow, UK
[8]Institute of Cardiovascular and Medical Sciences, University of Glasgow, Glasgow, UK
[9]School of Health and Life Sciences, Glasgow Caledonian University, Glasgow, UK
[10]Lancaster Medical School, Lancaster University, Lancaster, UK

**Acknowledgements** These data have been made available for analysis by members of the SCOTS Investigators group, Mr Afshin Alijani, Dr Ewan Bell, Professor Andrew Collier, Ms Carol Craig, Dr Jennifer Darrion, Mr David Galloway, Mr Hasan Kasem, Ms Joeleen McKean, Mr Stuart Oglesby, Mr Chris Shearer, Mr Rob Stuart and Mr Bruce Tulloh, due to their hard work in collecting data, and the patients who have participated in the study. The authors also wish to thank Mr Richard Welbourn, Ms Catherine Quinn, Professor David Beard, Dr John Mooney, Ms Elizabeth Thompson, Mrs Joanne O'Donnell, Mrs Jane Munro, Dr Iona Donnelly and Dr Samantha Alvarez Madrazo.

**Contributors** RMM, NG, AA, DB, JB, EG, ML, RL, NS, SS, IF and JL were involved in the conception, planning and design of the study, revising it critically and approving the final version for submission. NG conducted the analyses. RMM and JL wrote the first draft. All authors agree to be accountable for all aspects of the work and to ensure that questions relating to the accuracy or integrity of any part of the work are appropriately investigated and resolved.

**Funding** This work was supported by the National Institute for Health Research's Health Technology Assessment Programme (project number: 10/42/02).

**Disclaimer** The views expressed are those of the authors and not necessarily those of the National Institute for Health Research (NIHR) or the Department of Health and Social Care. Professor J Bruce is supported from NIHR Research Capability Funding via University Hospitals Coventry and Warwickshire.

**Competing interests** ML has received advisory board and speaker fees from Eli Lilly, Novo Nordisk, Merck and Sanofi. RL has received advisory board and speaker fees from Eli Lilly, Novo Nordisk and Servier. NS has consulted for Eli Lilly, Novo Nordisk, Pfizer and Sanofi and received compensation.

**Patient consent for publication** Not required.

**Ethics approval** A favourable ethical opinion for this study was obtained from the West of Scotland Research Ethics Committee 4 on 7 February 2013 (13/WS/0005).

**Provenance and peer review** Not commissioned; externally peer reviewed.

**Data availability statement** Data are available upon reasonable request. Deidentified participant data will be available upon reasonable request by bona fide researchers after completion of the study and our current funded analyses (estimated early 2022). Data requests should be submitted in the first instance to j.logue1@lancaster.ac.uk and will be reviewed by the study's publication committee.

of the translations (including but not limited to local regulations, clinical guidelines, terminology, drug names and drug dosages), and is not responsible for any error and/or omissions arising from translation and adaptation or otherwise.

**ORCID iDs**
Julie Bruce http://orcid.org/0000-0002-8462-7999
Mike Lean http://orcid.org/0000-0003-2216-0083
Naveed Sattar http://orcid.org/0000-0002-1604-2593
Sally Stewart http://orcid.org/0000-0002-9014-8601
Jennifer Logue http://orcid.org/0000-0001-9549-2738

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
