## [Reviewer comments · BMJ Open]

ARTICLE DETAILS

TITLE (PROVISIONAL)	Surgical Obesity Treatment Study (SCOTS), a prospective, observational cohort study on health and socio-economic burden in treatment-seeking individuals with severe obesity in Scotland, U.K.
AUTHORS	Mackenzie, Ruth; Greenlaw, Nicola; Ali, Abdulmajid; Bruce, Duff; Bruce, Julie; Grieve, Eleanor; Lean, Mike; Lindsay, Robert; Sattar, Naveed; Stewart, Sally; Ford, Ian; Logue, Jennifer

VERSION 1 – REVIEW

REVIEWER	Jeannot, Emilien Univ Appl Sci Western Switzerland
REVIEW RETURNED	03-Dec-2020

GENERAL COMMENTS	This article presents a secondary analysis of a larger SCOTS study, the article is clear and well constructed. All important elements are presented and explained. I have a few minor comments on the presentation of the article • Do we have any idea how long it took each participant to complete all these questionnaires?• Line 190: Why did you choose to stratify the BMI by 5 units?• In the version I received, the recruitment flow chart (figure 1) is completely illegible.• Given that for tables 2 and 3 you only present the data from the analyzed sample (n = 249), I am not sure of the interest of presenting the characteristics of the recruited sample in table 1.
---

REVIEWER	Sørum Falk, Ragnhild Research Support Services, Oslo University Hospital, Oslo Centre for Biostatistics and Epidemiology
REVIEW RETURNED	22-Dec-2020

GENERAL COMMENTS	It's a paper describing the SCOTS cohort, which aimed to estimate the association between health and socio-economic status with BMI and age. I congratulate the authors for using patient involvement so actively in the project. I have some comments and suggestions for improvement. Patient recruitment; in line 140 it is stated that patients 'were approached about the study at least four weeks prior to their primary bariatric surgery'. How could such large proportion of individuals (n=35) have surgery before baseline data? Due to the
---

	inclusion criteria 'over and undergoing their first bariatric surgery' (line 135)? What do "over" mean in this context? Questionnaires; it is an impressive number of validated Qs that was included in the study. Do you have some information regarding the time spent to complete the Qs? Do you think that this was one of the reasons for the drop-out rate? Regression analysis; when studying the association between BMI and outcome measures it seems fine to adjust for age as a confounder since age is associated with both BMI and the outcomes. However, when studying the association between age and the outcome measures it seems odd to adjust for BMI as BMI do not influence age. I find it useful to draw DAGs to aid the variable selections in the models and choice of method. Maybe mediation analysis or interaction analysis could be considered, depending on your question of interest? What is the correlation between age and BMI? Figure1; the reduction in individuals from 445 to 413 (n=32) don't match with n=1 in the 'out-going' box. Please add a box for the remaining 31 individuals. I suggest moving some of the information in lines 204-212 to this figure (e.g. add reasons in the 'did not return baseline questionnaire' box) Comparison of analyzed and required sample; I find it more relevant to compare the analyzed sample (n=249) with the required sample without PROM data available (444-249=195). As the analyzed sample is included in the required sample, the differences will be erased. I suggest adding an extra column to Table 1 with this information as well. Results section; the authors should be aware of the distinction between "increased or change" and "higher" risk. Studying or comparing separate groups of individuals the words "higher or lower" should be used when describing the differences; "change, increase and decrease" is reserved to changes within individuals over time. Eg. Lines 53, 266, 268, 270, 273, 275 etc. Please rephrase. Further, I suggest limiting to (none or) one digit when presenting figures in the text. Table 3; I find the set-up of table 3 a bit confusing. May switch rows (age and BMI) with columns (crude and adjusted)? Further, I think too much p-values are presented. Can an alternative be to present coefficients+CIs in bold if $p < 0.0001$? The restricted number of events in the binary outcomes should be stated as a limitation. The distinction between primary prevention (avoid severe obesity) and secondary prevention (treatment) should be clear Line 230 and line 257; replace cohort by sample Line 269; typo; OR 3.1 instead of 1.94? Write pre-operative or preoperative consistently
--	--

VERSION 1 – AUTHOR RESPONSE

Reviewer 1

Dr. Emilien Jeannot, Univ Appl Sci, Western Switzerland

Comment:

This article presents a secondary analysis of a larger SCOTS study, the article is clear and well constructed. All important elements are presented and explained. I have a few minor comments on the presentation of the article.

Response:

We thank Reviewer 1 for these supportive comments.

Comment:

Do we have any idea how long it took each participant to complete all these questionnaires?

Response:

We apologise for omitting this information from our manuscript and thank Reviewer 1 for bringing it to our attention. Prior to the study commencing, we piloted completion of the questionnaires with those patients involved in the design of the research. It took approximately 1 hour for the patients to complete the questionnaire. The estimated time it would take to complete each questionnaire (1 hour) was then stated clearly within the patient information leaflet and on the first page of all questionnaires sent to SCOTS participants during the course of the study. We have now included this information in the first paragraph of the 'Data Collection' sub-section of the 'PARTICIPANTS AND METHODS' section of the manuscript.

Comment:

Line 190: Why did you choose to stratify the BMI by 5 units?

Response:

We chose to stratify the BMI by 5 units in order to allow us to explore any subtleties in the effects of BMI in our cohort of individuals with severe obesity. There is obviously a big difference between a BMI of 41 and a BMI of 49 and so we stratified BMI by 5 units so that we could explore it in appropriate detail within our unique cohort of treatment-seeking individuals with severe obesity.

Comment:

In the version I received, the recruitment flow chart (figure 1) is completely illegible.

Response:

We are sorry to hear that Reviewer 1 was unable to view Figure 1. We can only assume that this was due to a technical problem at Reviewer 1's end as Reviewer 2 was able to view the figure and we have not been asked to make any formatting amendments to our manuscript. We hope Reviewer 1 will be able to see Figure 1 on receiving the amended version of our manuscript but please do let us know if you wish it to be submitted in a different format.

Comment:

Given that for tables 2 and 3 you only present the data from the analyzed sample (n = 249), I am not sure of the interest of presenting the characteristics of the recruited sample in table 1.

Response:

In presenting the characteristics of the recruited sample in Table 1, we are adhering to the STROBE (STrengthening the Reporting of OBServational studies in Epidemiology) Statement as outlined in the first paragraph of the 'PARTICIPANTS AND METHODS' section of our manuscript. Presenting the characteristics of the recruited sample in Table 1 allows us to demonstrate that our cohort (analysed sample) is representative of the recruited sample.

Reviewer 2

Dr. Ragnhild Sørnum Falk, Research Support Services, Oslo University Hospital

Comment:

It's a paper describing the SCOTS cohort, which aimed to estimate the association between health and socio-economic status with BMI and age. I congratulate the authors for using patient involvement so actively in the project. I have some comments and suggestions for improvement.

Response:

We thank Reviewer 2 for the supportive comments and the suggestions for improvement of our manuscript.

Comment:

Patient recruitment; in line 140 it is stated that patients 'were approached about the study at least four weeks prior to their primary bariatric surgery'. How could such large proportion of individuals (n=35) have surgery before baseline data?

Response:

All patients were indeed approached at least four weeks prior to their primary bariatric surgery. As outlined in paragraph 1 of the 'RESULTS' section of our manuscript, the reason that n=35 participants had surgery before baseline data was obtained, was that they didn't complete and return their baseline PROMs questionnaire to us in time/before their date of surgery. This was despite us sending them a number of reminders, as outlined in paragraph 1 of the 'Data Collection' sub-section of the 'PARTICIPANTS AND METHODS' section of our manuscript. The pre-surgery period can be a busy a stressful time for patients and that may have impacted.

Comment:

Due to the inclusion criteria 'over and undergoing their first bariatric surgery' (line 135)? What do "over" mean in this context?

Response:

We draw Reviewer 2's attention to the entire sentence which reads, "*Inclusion criteria were that patients were aged 16 years or over and undergoing their first bariatric surgery procedure.*" In this context, "over" means that patients were aged 16 years or older. We apologise for any confusion and have now added a comma so that the sentence reads as follows: "*Inclusion criteria were that patients were aged 16 years or over, and undergoing their first bariatric surgery procedure.*" We hope this is now clearer.

Comment:

Questionnaires; it is an impressive number of validated Qs that was included in the study. Do you have some information regarding the time spent to complete the Qs? Do you think that this was one of the reasons for the drop-out rate?

Response:

As per our response to Reviewer 1, we apologise for omitting this information from our manuscript and thank Reviewer 2 for bringing it to our attention. Prior to the study commencing, we piloted completion of the questionnaires with those patients involved in the design of the research. It took approximately 1 hour for the patients to complete the questionnaire. The estimated time it would take to complete each questionnaire (1 hour) was then stated clearly within the patient information leaflet and on the first page of all questionnaires sent to SCOTS participants during the course of the study. We have now included this information in the first paragraph of the 'Data Collection' sub-section of the 'PARTICIPANTS AND METHODS' section of the manuscript.

With regard to whether the time taken to complete the questionnaires was one of the reasons for participant drop-out, it may have been a factor, despite our extensive inclusion of patients in the design. In order to reflect this, we have now added the following sentence to the fifth paragraph of the 'DISCUSSION' section of our manuscript, "*The overall length of the questionnaire may have also played a role.*"

Comment:

Regression analysis; when studying the association between BMI and outcome measures it seems fine to adjust for age as a confounder since age is associated with both BMI and the outcomes. However, when studying the association between age and the outcome measures it seems odd to adjust for BMI as BMI do not influence age. I find it useful to draw DAGs to aid the variable selections in the models and choice of method. Maybe mediation analysis or interaction analysis could be considered, depending on your question of interest? What is the correlation between age and BMI?

Response:

We thank Reviewer 2 for this comment but wish to clarify that we are not suggesting that BMI and age influence each other. Rather, we are looking at their independent effects. We are looking for the association between BMI and each QoL indicator/functional measure and, independently, also the association between age and each QoL indicator/functional measure. Given we anticipate that BMI may have an association with these QoL indicator/functional measures, and age may also have an association with these QoL indicator/functional measures, our data allows us to explore these associations independently in unadjusted models and then in one combined model after further adjusting for other confounding data that may each also have an association with these QoL/functional measures.

We performed these analyses because our sample varies from the international cohorts by having participants with both higher BMI and older age, both of which are an indication that surgery is left until later in the disease process. As such, we wanted to know what effect BMI and age each had independently on QoL in the cohort as these are novel data, certainly for the U.K. population. We performed the analyses in this way to be able to make recommendations that may have an effect on clinical policy.

Comment:

Figure1; the reduction in individuals from 445 to 413 (n=32) don't match with n=1 in the 'out-going' box. Please add a box for the remaining 31 individuals. I suggest moving some of the information in lines 204-212 to this figure (e.g. add reasons in the 'did not return baseline questionnaire' box)

Response:

We apologise if Figure 1 was not clear. We have now added an additional box to the figure to show that n=31 patients consented to only part 1 of the study (data linkage). We hope that this, together with the explanatory information given in both the figure legend and in paragraph 1 of the 'RESULTS' section of the manuscript, improves clarity.

With regard to adding reasons for questionnaires not being returned to Figure 1, we cannot do this as we do not know why they did not return the questionnaire. We speculate in the discussion that non-progression to surgery may have played a part, but this is not universal as some who did not progress did return the questionnaire.

Comment:

Comparison of analyzed and recruited sample; I find it more relevant to compare the analyzed sample (n=249) with the required sample without PROM data available (444-249=195). As the analyzed sample is included in the required sample, the differences will be erased. I suggest adding an extra column to Table 1 with this information as well.

Response:

We thank Reviewer 2 for this comment but wish to highlight the fact that there is no statistically significant differences between those we are including in our analysis population and those we are not including. We draw Reviewer 2's attention to the table included below which summarises the data available (sex, age, BMI and SIMD) in all 444 consenting patients: those that we were including for analysis (n = 249) and those that either did not consent to PROMS, or did not complete their baseline PROMS prior to operation. The p-values included compare the mutually exclusive groups - the analysable with the non-analysable, using the same methods as detailed in the methods that we used for the comparisons between age groups/BMI groups, etc. We can see that there are no statistically

significant differences between those that we are including in our analysis population and those that we are not including.

Characteristic		Recruited Sample	Analysed Sample*	Non-analysable**	p-value***
		N = 444	N = 249	N = 195	
Sex, N (%)	Male	123 (27.7)	72 (28.9)	51 (26.2)	0.5187
	Female	321 (72.3)	177 (71.1)	144 (73.8)	
	Missing	0	0	0	
Age (years)	Mean (SD)	46.2 (9.1)	45.9 (9.1)	46.4 (9.1)	0.5489
	Missing	0	0	0	
Age Group, N (%)	< 35 years	61 (13.7)	36 (14.5)	25 (12.8)	0.9649
	35-44 years	116 (26.1)	63 (25.3)	53 (27.2)	
	45-49 years	109 (24.5)	63 (25.3)	46 (23.6)	
	50-54 years	79 (17.8)	43 (17.3)	36 (18.5)	
	55+ years	79 (17.8)	44 (17.7)	35 (17.9)	
	Missing	0	0	0	
BMI (kg/m ²)	Median (Q1, Q3)	47.2 (42.7 ; 53.6)	47.6 (42.8 ; 53.8)	46.3 (42.4 ; 52.9)	0.1754
	Missing	1	0	1	
BMI Group, N (%)	BMI < 40	52 (11.7)	24 (9.6)	28 (14.4)	0.4684
	BMI 40-44	115 (26.0)	64 (25.7)	51 (26.3)	
	BMI 45-49	116 (26.2)	64 (25.7)	52 (26.8)	
	BMI 50-54	71 (16.0)	44 (17.7)	27 (13.9)	
	BMI 55+	89 (20.1)	53 (21.3)	36 (18.6)	
	Missing	1	0	1	
SIMD Quintile, N (%)	Quintile 1 (most deprived)	135 (30.5)	70 (28.3)	65 (33.3)	0.4783
	Quintile 2	108 (24.4)	65 (26.3)	43 (22.1)	
	Quintile 3	84 (19.0)	51 (20.6)	33 (16.9)	
	Quintile 4	68 (15.4)	34 (13.8)	34 (17.4)	
	Quintile 5 (least deprived)	47 (10.6)	27 (10.9)	20 (10.3)	
	Missing	2	2	0	

*Participants who returned baseline questionnaires prior to their bariatric surgery are included in the analysed sample.

**Participants who either did not consent to PROMS or did not return their baseline questionnaires prior to their bariatric surgery are included in the non-analysable sample.

***p-value of comparisons between the analysed and non-analysable samples.

In order to make this clearer in the manuscript, we have added the following sentence to the 'Characteristics of recruited and analysed sample' sub-section of the 'RESULTS' section of our manuscript, "There were no statistically significant differences between the analysed sub-set (n=249) and the non-analysed sub-set (n=195)."

Comment:

Results section; the authors should be aware of the distinction between “increased or change” and “higher” risk. Studying or comparing separate groups of individuals the words “higher or lower” should be used when describing the differences; “change, increase and decrease” is reserved to changes within individuals over time. Eg. Lines 53, 266, 268, 270, 273, 275 etc. Please rephrase.

Response:

We apologise for these oversights and thank Reviewer 2 for bringing them to our attention. We have now amended the wording as required.

Comment:

Further, I suggest limiting to (none or) one digit when presenting figures in the text.

Response:

We appreciate this point with regard to decimal places and have now altered our manuscript to ensure that all values in the text are given to only one decimal place (with the exception of p values).

Comment:

Table 3; I find the set-up of table 3 a bit confusing. May switch rows (age and BMI) with columns (crude and adjusted)? Further, I think too much p-values are presented. Can an alternative be to present coefficients+CI in bold if $p < 0.0001$?

Response:

We apologise if Table 3 was confusing and have now reformatted it in an attempt to make it easier to understand. The table now has fewer columns and we have removed p-values as it should be evident from the confidence interval whether the result has a significant p-value or not (at the 5% level).

Comment:

The restricted number of events in the binary outcomes should be stated as a limitation.

Response:

We apologise for this oversight and have now added the following sentence to the fifth paragraph of the ‘DISCUSSION’ section of our manuscript, “*Low numbers of some comorbidities meant that this could not be a focus of this analysis.*”

Comment:

The distinction between primary prevention (avoid severe obesity) and secondary prevention (treatment) should be clear

Response:

We apologise for the lack of clarity here and have amended the final paragraph of our ‘DISCUSSION’ section to read, “*While primary prevention of obesity is clearly paramount to avoid more people developing such a debilitating, chronic condition, investment is urgently needed.....*”. We hope this makes the distinction between primary and secondary prevention clearer.

Comment:

Line 230 and line 257; replace cohort by sample

Response:

We have now replaced cohort with sample in these lines of the manuscript.

Comment:

Line 269; typo; OR 3.1 instead of 1.94?

Response:

We apologise for this mistake and thank Reviewer 2 for bringing it to our attention. We have now changed OR from 1.94 to 3.1 in the first paragraph of the 'Comorbidity by BMI and age' subsection of the 'RESULTS' section.

Comment:

Write pre-operative or preoperative consistently

Response:

We apologise for this inconsistency and have now used "preoperative" throughout the manuscript.

VERSION 2 – REVIEW

REVIEWER	Sørum Falk, Ragnhild Research Support Services, Oslo University Hospital, Oslo Centre for Biostatistics and Epidemiology
REVIEW RETURNED	24-Feb-2021

GENERAL COMMENTS	Thanks for all the changes that have been performed. I think the manuscript has improved a lot. However, I have some further issues regarding the regression analysis that needs to be clarified. When performing regression analysis, one have to carefully consider which (and how) to adjust for potential confounding factors. Although looking at independent associations, I think it's important to have the causal diagram in mind, and to think of how the different variables are influencing each other. From Table 3, we see that estimated coefficients are similar in crude and adjusted analysis, which may points towards independent associations for age and BMI. It is however relevant to know the correlation between age and BMI, as both are considered the main 'exposures' of interest. What is the correlation? Further, by adjusting for e.g. smoke when studying the effect of BMI on QoL, the underlying assumption is that the linear effect of BMI has the same slope for both smokers and non- smokers. Is this the case? As previously pointed out, interaction analysis could be considered. A final suggestion is to add "data not shown" after the sentence "There were no statistically significant differences between the analysed sub-set (n=249) and the non-analysed sub-set (n=195)." or eventually present data in supplement.
--

VERSION 2 – AUTHOR RESPONSE**Response to Reviewer**

Reviewer: 2

Dr. Ragnhild Sørum Falk, Research Support Services, Oslo University Hospital

Comments to the Author:

Comment:

Thanks for all the changes that have been performed. I think the manuscript has improved a lot. However, I have some further issues regarding the regression analysis that needs to be clarified.

Response:

Thank you very much for this comment on the improvement of our manuscript.

Comment:

When performing regression analysis, one have to carefully consider which (and how) to adjust for potential confounding factors. Although looking at independent associations, I think it's important to have the causal diagram in mind, and to think of how the different variables are influencing each other. From Table 3, we see that estimated coefficients are similar in crude and adjusted analysis, which may points towards independent associations for age and BMI. It is however relevant to know the correlation between age and BMI, as both are considered the main 'exposures' of interest. What is the correlation?

Further, by adjusting for e.g. smoke when studying the effect of BMI on QoL, the underlying assumption is that the linear effect of BMI has the same slope for both smokers and non- smokers. Is this the case? As previously pointed out, interaction analysis could be considered.

Response:

There was no significant correlation between age and BMI (correlation = 0.0075, p=0.9061). This information has now been added to the 'Comorbidity by BMI and age' sub-section of the 'Results' section of the manuscript. We also remind the reviewer that the associations involving age and BMI separately, compared to those where both age and BMI were included in the models, were qualitatively similar.

We have now added interaction analyses focused on smoking as per the reviewer's request. Significant interactions with smoking have been added to the 'Comorbidity by BMI and age' sub-section of the 'Results' section of the manuscript and included as Supplementary Table 4. To summarise, there was a borderline significant interaction between age and smoking for the outcome 'use of any aids or specialist equipment' and a significant interaction between BMI and smoking for the outcome PHQ-9 (binary).

Comment:

A final suggestion is to add "data not shown" after the sentence "There were no statistically significant differences between the analysed sub-set (n=249) and the non-analysed sub-set (n=195)." or eventually present data in supplement.

Response:

We thank the reviewer for pointing out this oversight and "data not shown" has now been added to the manuscript where requested.

VERSION 3 – REVIEW

REVIEWER	Sørum Falk, Ragnhild Research Support Services, Oslo University Hospital, Oslo Centre for Biostatistics and Epidemiology
REVIEW RETURNED	29-Jun-2021

GENERAL COMMENTS

I have no further comments except that (one) figures should be dobbled checked.
In the Main_document_-_Marked_copy in Line 281 /table 3, the OR of Age on use of aids 3.1. In another version of the manuscript, the OR is 3.4.